# On the phase dependence of the soliton collisions in the Dyachenko-Zakharov envelope equation.

Dmitry Kachulin[1] and Andrey Gelash[1,2]

[1]Novosibirsk State University, Novosibirsk, 630090, Russia
[2]Institute of Thermophysics, SB RAS, Novosibirsk, 630090, Russia

**Correspondence:** Dmitry Kachulin, d.kachulin@gmail.com

**Abstract.**

We study soliton collisions in the Dyachenko-Zakharov equation for the envelope of gravity waves in deep water. The numerical simulations of the soliton interactions revealed several fundamentally different effects when compared to analytical two-soliton solutions of the nonlinear Schrodinger equation. The relative phase of the solitons is shown to be the key parameter determining the dynamics of the interaction. We find that the maximum of the wave field can significantly exceed the sum of the soliton amplitudes. The solitons lose up to a few percent of their energy during the collisions due to radiation of incoherent waves and in addition exchange energy with each other. The level of the energy loss increases with certain synchronisation of soliton phases. Each of the solitons can gain or lose the energy after collision resulting in the increase or decrease of the amplitude. The magnitude of the space shifts that solitons acquire after collisions depends on the relative phase and can be either positive or negative.

## 1 Introduction

The existence and interactions of coherent structures like solitons and breathers on the surface of a deep water are a remarkably rich and fascinating subject for both experimental and theoretical studies. The exact mathematical model describing gravity waves in the ocean is the Euler equation, yet it is often rather complicated to study it by analytic or numerical means. Instead various reduced models for water waves have demonstrated good agreement with the experimental data and have been widely adopted in fluid dynamics and geophysics communities.

The most prominent and widely used model for weakly nonlinear surface waves in deep water is the nonlinear Schrödinger equation (NLS). It describes time evolution of the envelope of a quasi–monochromatic wave train (Zakharov (1968)) and is integrable via the inverse scattering transform (IST) in 1D (Zakharov and Shabat (1972)). Other models for weakly nonlinear waves include the Dysthe equation (Dysthe (1979)), and the compact Dyachenko–Zakharov equation (DZ) (Dyachenko and Zakharov (2011)) neither of which is known to be integrable by the IST.

By means of the IST one can find NLS soliton solutions and track their evolution in time until their collision and beyond analytically. The collision of the NLS solitons is perfectly elastic, that is no loss of the energy occurs. The equations which are not integrable by the IST may have exact stationary solitary solutions interacting inelastically. For example the Dysthe

equation is known to admit solitary solutions whose existence has been demonstrated by other approaches unrelated to the IST (see Akylas (1989); Zakharov and Dyachenko (2010)).

Both the NLS and the Dysthe equations are formulated to describe the evolution of the envelope function. They require that the steepness of the wave train is small and it is modulated weakly, i.e., there are sufficiently many carrier wave lengths in the characteristic scale of the modulation. In terms of the Fourier transform of the surface elevation this is equivalent to having a sufficiently narrow band concentrated in the vicinity the carrier wave number. The DZ equation is formulated for the wave train itself and is free from the assumptions of the weak nonlinearity and narrow bandness (Dyachenko and Zakharov (2011, 2012)). More precisely the DZ equation describes the evolution of the surface displacement and the surface velocity potential in terms of special canonical variables discussed below. The solitary type solutions to the DZ equation are commonly referred to as the breather solutions, or simply breathers. The DZ breathers are found numerically and their interaction has been the subject of the works Fedele and Dutykh (2012a, b); Dyachenko et al. (2013). The following work by Fedele (2014) investigated the properties of the DZ equation for various values of wave steepness. In particular it was shown that the dynamics of the DZ equation becomes of a modified Korteweg–de Vries (mKdV) equation type when the value of steepness is large enough providing a possible mechanism of wave breaking.

In the work Zakharov et al. (2006) solitons of the NLS equation were found to be a fair model for propagating solitary wave groups in the Euler equation at small steepness. The strongly nonlinear breather solutions of the Euler equation were found numerically in Dyachenko and Zakharov (2008), and subsequent works Slunyaev (2009); Slunyaev et al. (2013, 2017) study propagation and interaction of these breathers numerically and in water tank experiments.

The study of soliton (or breather) interactions in the reduced deep water models is an important step in understanding of the surface waves dynamics and the fundamental properties of the Euler equation. In this work we focus on the DZ equation in the form suggested by Dyachenko et al. (2017a) which describes the wave train envelope without any assumptions on its spectral width. Hereafter we refer this equation to as the Dyachenko–Zakharov envelope equation, or the DZe equation. The envelope form of the DZ equation allows a direct comparison with the more restricted but the more established integrable NLS equation. In this work we always use the term "solitons” to describe the envelope solutions of the NLS equation, the Dysthe equation and the DZe equation; we also refer to the solitary solutions of the DZ equation and the Euler equation as "breathers” when we imply wave train itself rather than its envelope.

The soliton interactions in the NLS equation depend drastically on their relative complex phases, e.g. the maximum amplification of the amplitude in a collision is determined by the synchronization of the phases of the solitons. The phase synchronization plays an important role in the formation of the waves of extreme amplitude, the rogue waves, and has been studied in the water wave theory (Kharif et al. (2009)) as well as in other contexts like optical pulses in fibre (Antikainen et al. (2012)). In the recent works Sun (2016) and Gelash (2018) phase synchronization in multisoliton ensembles has been studied analytically. The role of the soliton phase parameters has been extensively studied for other integrable models including mKdV equation for long waves (Slunyaev and Pelinovsky (2016)).

In the present work we study soliton interactions in the DZe equation and its dependence on the phases of interacting solitons. We demonstrate how the amplitude amplification, the energy exchange between the solitons, the energy loss to emission of incoherent radiation and the space shift of the solitons after collisions reveal fundamental differences from the NLS equation.

## 2    The envelope equations for deep water gravity waves

A one-dimensional potential flow of an ideal fluid of infinite depth in presence of gravity is a Hamiltonian system. The surface elevation $\eta(x,t)$ and the velocity potential $\psi(x,t)$ at the surface are canonically conjugated variables (Zakharov (1968)).

Dyachenko and Zakharov (2011, 2012) suggested a canonical transformation from the physical real–valued Hamiltonian variables $\eta(x,t)$ and $\psi(x,t)$ to the complex normal variable $b(x,t)$. The DZ equation is found by taking a fourth order expansion of the Hamiltonian in powers of $|b(x,t)|$ and assuming that all waves are propagating in a single direction. Recently Dyachenko et al. (2016a, 2017b) introduced the new canonical variable $c(x,t)$, such that the DZ equation can be written in $x$–space in the following "super" compact form:

$$\frac{\partial c}{\partial t} + i\hat{\omega}_k c - i\partial_x^+ \left( |c|^2 \frac{\partial c}{\partial x} \right) = \partial_x^+ \left( \hat{k}(|c|^2)c \right). \tag{1}$$

Here $g$ is the free–fall acceleration and the operators $\hat{k}$ and $\hat{\omega}$ are Fourier multipliers by the wavenumber $|k|$ and the linear wave frequency $\sqrt{g|k|}$ respectively. The operator $\partial_x^+$ in the Fourier space is $ik\theta(k)$, where $\theta(k)$ is the Heaviside step function. The physical variables, $\eta$ and $\psi$ can be recovered aposteriori by the canonical transformation. The surface elevation $\eta(x,t)$ to the order $|c|^2$ is the following:

$$\eta(x,t) = \frac{1}{\sqrt{2}g^{\frac{1}{4}}} (\hat{k}^{-\frac{1}{4}} c(x,t) + \hat{k}^{-\frac{1}{4}} c(x,t)^*) + \frac{\hat{k}}{4\sqrt{g}} \left[ \hat{k}^{-\frac{1}{4}} c(x,t) - \hat{k}^{-\frac{1}{4}} c(x,t)^* \right]^2, \tag{2}$$

where the operator $\hat{k}^\alpha$ is a Fourier multiplier by $|k|^\alpha$, and star denotes a complex conjugate quantity.

The equation (1) has a breather solution:

$$c(x,t) = c_{br}(x - Vt)e^{i(\tilde{k}x - \tilde{\omega}t)}, \tag{3}$$

where $\tilde{k}$ is the carrier wavenumber, $V = \frac{1}{2}\sqrt{g/\tilde{k}}$ is the group velocity in the laboratory frame of reference and $\tilde{\omega}$ is a nonlinear frequency close to $\sqrt{g\tilde{k}}$. In Fourier space this solution has the following form:

$$c_k(t) = \frac{1}{\sqrt{2\pi}} \int c_{br}(x - Vt)e^{i(\tilde{k}-k)x}e^{-i\tilde{\omega}t}dx = \frac{1}{\sqrt{2\pi}} \int c_{br}(\xi)e^{i(\tilde{k}-k)\xi}e^{-i(\tilde{\omega}-\tilde{k}V+kV)t}d\xi = \varphi_k e^{-i(\Omega+Vk)t}, \tag{4}$$

where

$$\varphi_k = \frac{1}{\sqrt{2\pi}} \int c_{br}(\xi)e^{i(\tilde{k}-k)\xi}d\xi. \tag{5}$$

In formula (4) instead of $\tilde{\omega}$ we use the new frequency parameter $\Omega$:

$$\Omega = \tilde{\omega} - \tilde{k}V = \tilde{\omega} - \frac{\sqrt{g\tilde{k}}}{2}. \tag{6}$$

Breather solutions can be found numerically by Petviashvili method (Petviashvili (1976)) and the details are given in (Dyachenko et al. (2017b)). The solution $\varphi_k$ can be found numerically by iterations:

$$5 \quad \varphi_k^{(n+1)} \quad = \quad \frac{NL_k^{(n)}}{M_k} \left[ \frac{\sum_{k'}(\varphi_{k'}^{(n)} NL_{k'}^{(n)})}{\sum_{k'}(\varphi_{k'}^{(n)} M_{k'} \varphi_{k'}^{(n)})} \right]^{-\frac{3}{2}}. \tag{7}$$

Here $\varphi_k^{(n)}$ is the breather solution $\varphi_k$ on the $n$-th iteration,

$$M_k = \Omega + Vk - \omega_k. \tag{8}$$

The symbol $NL^{(n)}$ denotes the nonlinear part of the equation (1) on the $n$-th iteration in the $x$-space:

$$NL^{(n)} \quad = \quad -\frac{\partial^+}{\partial x}\left(|\varphi^{(n)}|^2 \frac{\partial \varphi^{(n)}}{\partial x}\right) + i\frac{\partial^+}{\partial x}\left(\hat{k}\left(|\varphi^{(n)}|^2\right)\varphi^{(n)}\right), \tag{9}$$

and $NL_k^{(n)}$ is the discrete Fourier transform of $NL^{(n)}$. The breather solution is determined by two independent parameters: the group velocity $V$ and the frequency $\Omega$. The value of the first parameter $V = \frac{1}{2}\sqrt{g/\tilde{k}}$ defines the carrier wave number $\tilde{k}$ (and the carrier wave length $\lambda = 2\pi/\tilde{k}$) of the solitary group. The second parameter $\Omega$ has the value close to $\frac{1}{2}\sqrt{g\tilde{k}}$ (or $g/4V$, see formula (6)) and implicitly defines the shape and the amplitude of the breather. The breather solutions found by Petviashvili method (7) are determined up to an arbitrary phase factor $e^{i\phi}$.

Recently Dyachenko et al. (2017a) derived the envelope version of the super compact equation (1) using the envelope function $C(x,t)$:

$$c(x,t) = C(x,t)e^{i(k_0 x - \omega_{k_0}t)}, \tag{10}$$

where $k_0$ is an arbitrary characteristic wavenumber and $\omega_{k_0} = \sqrt{gk_0}$ is the corresponding linear frequency. The Dyachenko-Zakharov envelope (DZe) equation written in the reference frame moving with the group velocity $V_0 = \frac{\partial \omega}{\partial k}|_{k_0} = \frac{\omega_{k_0}}{2k_0}$ has the

following form:

$$\frac{\partial C}{\partial t} \quad + \quad i\left[\omega_{k_0+k} - \omega_{k_0} - \frac{\partial \omega_{k_0}}{\partial k_0}k\right]\hat{\theta}_{k_0+k}C + ik_0^2\hat{\theta}_{k_0+k}\left[|C|^2 C\right] +$$
$$+ \quad k_0\hat{\theta}_{k_0+k}\left[C\frac{\partial}{\partial x}|C|^2 + 2|C|^2\frac{\partial C}{\partial x} - i\hat{k}(|C|^2)C\right] -$$
$$- \quad \hat{\theta}_{k_0+k}\frac{\partial}{\partial x}\left[\hat{k}(|C|^2)C + i|C|^2\frac{\partial C}{\partial x}\right] = 0. \tag{11}$$

The DZe equation (11) is Hamiltonian, and the Hamiltonian is

$$\mathcal{H} \quad = \quad \int C^*\hat{V}_k C\, dx + \frac{1}{2}\int |C|^2\left[k_0|C|^2 + \frac{i}{2}(CC'^* - C^*C') - \hat{k}|C|^2\right]dx, \tag{12}$$

where the operator $\hat{V}_k$ has the following form in $k$–space:

$$V_k = \frac{\left[\omega_{k_0+k} - \omega_{k_0} - \frac{\partial \omega_{k_0}}{\partial k_0} k\right]}{k_0 + k}. \tag{13}$$

The DZe equation was derived without any assumptions on the spectral width of the wave packet. Moreover, the equation (11) is the exact envelope form of the equation (1) and has the same range of applicability. Solutions of the equation (11) and the equation (1) are linked by the transformation (10). The solutions (3) written in terms of the envelope function $C(x,t)$ has the following soliton form:

$$C_{br}(x,t) = c_{br}(x - Vt) e^{i(\tilde{k} - k_0)x - i(\tilde{\omega} - \omega_{k_0})t}. \tag{14}$$

The DZe equation (11) reduces to the Dysthe equation and further to the NLS equation as the Fourier spectrum of $c(x,t)$ becomes increasingly localized at $k_0$.

In this work we study only the model (11) itself and the NLS equation which can be extracted from (11) as:

$$\frac{\partial C}{\partial t} + \frac{i\omega_{k_0}}{8k_0^2} \frac{\partial^2 C}{\partial x^2} + ik_0^2 \left[|C|^2 C\right] = 0. \tag{15}$$

Soliton solutions of equations (11) and (15) will be compared in the next sections.

## 3   Soliton solutions of the NLS equation and the DZe equation

We consider solitons in the frame moving with the velocity $V_0 = \frac{\partial \omega}{\partial k}|_{k_0} = \frac{\omega_{k_0}}{2k_0}$. The one-soliton solution of the NLS equation (15) moving in the frame with velocity $U$ can be written as:

$$C_s(x,t) = C_0 \operatorname{sech}\left[\frac{2C_0 k_0^2}{\sqrt{\omega_{k_0}}} ((x - x_0) - Ut)\right] \exp\left[-i\frac{4k_0^2}{\omega_{k_0}} U(x - x_0) + i\frac{2U^2 k_0^2}{\omega_{k_0}} t - i\frac{C_0^2 k_0^2}{2} t + i\phi_0\right], \tag{16}$$

where $C_0$ is the soliton amplitude, $x_0$ is the soliton location at $t = 0$, $\phi_0$ is an arbitrary soliton phase. The shape and width of NLS soliton for a fixed wavenumber $k_0$ (and velocity $V_0$) is defined by a single independent parameter $C_0$.

In this work we focus on the interactions of the NLS solitons and the DZe solitons of equal amplitude $C_0$ and various velocities $V = V_0 + U$. To describe soliton collisions using the NLS model analytically we hold the carrier wave number $k_0$ fixed and vary the relative velocity $U$. Thus in our studies all the NLS solitons have the same modulus $|C_s(x)|$.

The dynamics of DZe solitons collisions can be investigated only by numerical simulations. We study interactions of the DZe solitons of the same amplitudes $C_0$ and different velocities $V = V_0 + U$, like in the case of the NLS solitons. The amplitude of the DZe soliton $C_0$ is uniquely determined by parameters $V$ and $\Omega$ of the Petviashvili method. However, there are no analytical relations between soliton amplitude, $V$ and $\Omega$. Therefore to find the DZe soliton having the given velocity $V$ and amplitude $C_0$ we fix $V$ and vary parameter $\Omega$ (6) in the Petviashvili method. The shape of the soliton solutions of DZe equation found in this way differ from each other. More precisely, these solitons have different characteristic widths (see the curves 1,2 and 3 in figure 1). When $V = V_0$ the soliton solution (14) of the DZe equation almost coincides with the NLS soliton (16) – see the curves

3 and 4 in figure 1. The soliton solution (14) with $V > V_0$ is the envelope of the wave group having the carrier wave number $\tilde{k} < k_0$ while in the case $V < V_0$ the carrier wave number $\tilde{k} > k_0$. The characteristic steepness of the wave group of amplitude $C_0$ is proportional to $\tilde{k}^{3/4} C_0$ (see formula (2)). Thus the wave group with $V < V_0$ is steeper (and have higher nonlinearity) than the wave groups with $V = V_0$ and $V > V_0$. The steeper waves need stronger dispersion to balance the solitary wave group, and hence the soliton represented by the curve 2 is shorter than solitons represented by the curves 3 and 1.

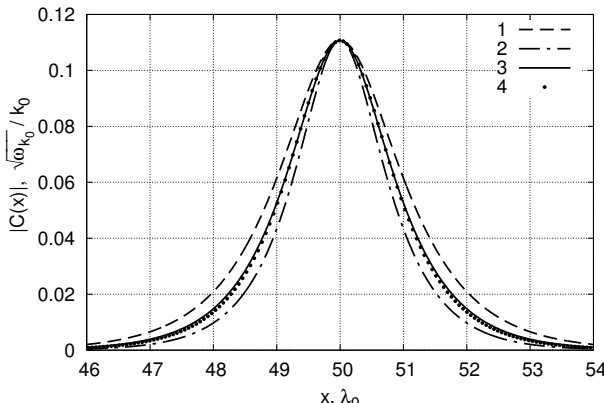

**Figure 1.** Comparison of DZe solitons and NLS soliton having amplitude $C_0 = 1.11 \cdot 10^{-1} \sqrt{\omega_{k_0}}/k_0$. The curves show the absolute value of the envelope function $|C(x)|$ for the DZe solitons with parameters: $U = 0.04\,V_0$, $\Omega_1 = 4.86 \cdot 10^{-1}\,\omega_{k_0}$ (dashed curve 1); $U = -0.04\,V_0$, $\Omega_2 = 5.28 \cdot 10^{-1}\,\omega_{k_0}$ (dash-doted curve 2); $U = 0$, $\Omega_0 = 5.06 \cdot 10^{-1}\,\omega_{k_0}$ (solid curve 3). The dots 4 show the absolute value of the envelope function for the NLS soliton with $U = 0$.

## 4  The interactions of the solitons

We fix a carrier wave number $k_0$ for the DZe (11) and for the NLS model (15), i.e. consider the dynamics of solitons in a frame moving with the velocity $V_0 = \frac{\partial \omega}{\partial k}|_{k_0} = \frac{\omega_{k_0}}{2k_0}$. We study interactions of two solitons having (in the laboratory reference frame) close unidirectional velocities $V = V_0 + U_0$ and $V = V_0 - U_0$. We compare four cases of two–soliton interactions that correspond to four values of the maximum wave steepness $\mu$ (and amplitudes $C_0$ correspondingly):

- $\mu \approx 0.05$ (amplitude $C_0 = 3.16 \cdot 10^{-2}\,\frac{\sqrt{\omega_{k_0}}}{k_0}$),

- $\mu \approx 0.1$ (amplitude $C_0 = 7.12 \cdot 10^{-2}\,\frac{\sqrt{\omega_{k_0}}}{k_0}$),

- $\mu \approx 0.15$ (amplitude $C_0 = 8.85 \cdot 10^{-2}\,\frac{\sqrt{\omega_{k_0}}}{k_0}$),

- $\mu \approx 0.2$ (amplitude $C_0 = 1.11 \cdot 10^{-1}\,\frac{\sqrt{\omega_{k_0}}}{k_0}$).

The steepness $\mu$ is determined as the maximum of the derivative of the surface elevation:

$$\mu = \max |\eta'(x)|,$$

and $\eta(x)$ is recovered from the transformation (2). The dimensionless wave steepness $\mu \sim C_0 \frac{k_0}{\sqrt{\omega_{k_0}}}$ (see formula (2)), that is why we measure the wave field amplitude $C(x)$ in the units $\frac{\sqrt{\omega_{k_0}}}{k_0}$. For each case the size of the computational domain was $x/\lambda_0 \in [0,100]$ where $\lambda_0 = 2\pi/k_0$. The relative velocity was $U_0 = 0.04V_0$ and at the initial time the solitons are located at $x = 25\lambda_0$ and $x = 75\lambda_0$. For the sake of brevity we label the soliton that was initially located at $25\lambda_0$ and the other soliton by

the indices 1 and 2 respectively. The total simulation time is $50\lambda_0/U_0 = 2500 T_0$, where $T_0 = 2\pi/\omega_{k_0}$ is the time period for the base wave number $k_0$.

The NLS equation is a completely integrable model and exact multisoliton solution is available (see the work Zakharov and Shabat (1972)). We use this analytic solution to study the collision of solitons for the NLS case. In figure 2 we present an example of interacting solitons and illustrate how their collision leads to a space shift in the positions of the solitons as well as

the formation of a nonlinear wave profile with a peak amplitude $2C_0$. In the NLS model the space shift $\delta x$ is determined by the soliton amplitudes and velocities and it does not depend on the phase. Each soliton acquires a positive shift in the direction of its propagation (as we mentioned above we consider the system of reference moving with velocity $V_0$ where the solitons propagate in different directions) and is calculated from the formula (see Novikov et al. (1984)):

$$\delta x = \frac{\omega_{k_0}}{2C_0 k_0^2} \lg\left(1 + \frac{\omega_{k_0}}{4}\left(\frac{C_0}{U_0}\right)^2\right). \tag{17}$$

In the case illustrated in the figure 2 $\delta x = 1.55\,\lambda_0$. In addition to the space shift (17) the solitons acquired a phase shift $\delta\phi$ that is calculated using similar expression:

$$\delta\phi = \arg\left(1 - i\frac{\sqrt{\omega_{k_0}}}{2}\frac{C_0}{U_0}\right). \tag{18}$$

As one can see from (16) the dependence of the soliton phase at its center on time is:

$$\phi(t) = \phi_0 - \left(\frac{2U^2 k_0^2}{\omega_{k_0}} + \frac{C_0^2 k_0^2}{2}\right)t. \tag{19}$$

Thus for the two solitons of equal amplitudes $C_0$ and the relative velocities $\pm U_0$ the phase difference is time invariant: $\Delta\phi(t) \equiv \phi_{02} - \phi_{01}$, where $\phi_{01}$ and $\phi_{02}$ are the initial phases of the soliton 1 and 2 respectively. For the case of the NLS solitons of equal amplitudes the space and the phase shifts given by equations (17) and (18) are mutually compensated.

The maximal amplitude $2C_0$ is achieved when the phase difference between the colliding NLS solitons is equal to zero: $\Delta\phi = 0$ (see e.g. Antikainen et al. (2012)). The value of the maximum amplitude amplification depends on the relative phase

of the interacting solitons $\Delta\phi$. We use the normalized definition of the maximum amplitude amplification function $A(\Delta\phi)$ given by:

$$A(\Delta\phi) = \left.\frac{\max\limits_{(x,t)}\left(|C(x,t)|\right)}{2C_0}\right|_{\Delta\phi}. \tag{20}$$

In other words we find maximum amplitude of the wave field formed during the whole collision process and normalise it to the sum of the soliton amplitudes.

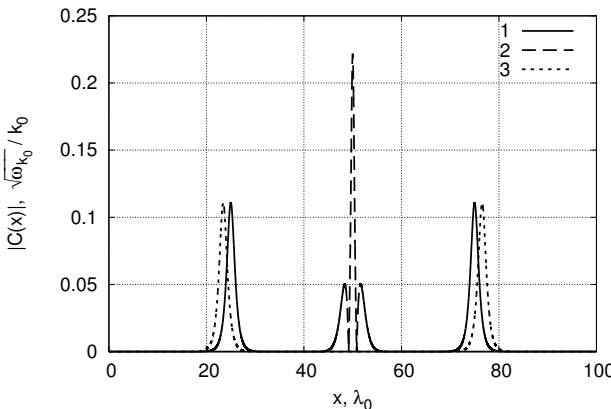

**Figure 2.** Collision of NLS solitons with amplitudes $C_0 = 1.11 \cdot 10^{-1} \sqrt{\omega_{k_0}}/k_0$, velocities $U = \pm 0.04\,V_0$ and phase difference $\Delta\phi = 0$. The curves show the absolute value of the envelope function $|C(x)|$ at the moment of time: $t = 0$ (solid curve 1); at the moment of maximum amplitude amplification during the collision process (dashed curve 2) and the moment of time $t = 50\frac{\lambda_0}{U_0} = 2500\,T_0$ (dotted curve 3), i.e. after soliton collision.

In the NLS model the amplitude amplification function decreases when the $|\Delta\phi|$ grows and achieves its minimum at $\Delta\phi = \pm\pi$ (see figure 3). In a more general case of the collision of NLS solitons of unequal amplitudes the phase difference $\Delta\phi$ is time dependent. In such a case we must choose a time $t_c$ when $\Delta\phi$ is defined. We choose $t_c = 25\,\lambda_0/U_0 = 1250\,T_0$ which is the time when either of the solitons reaches the center of the domain in the absence of the other. In this case the amplitude amplification $A(\Delta\phi)$ is similar to the amplitude amplification presented in the figure 3 with the exception that the maximum is shifted from $\Delta\phi = 0$. This is caused by unequal values of the soliton space shifts $\delta x_1$ and $\delta x_2$, and phase shifts $\delta\phi_1$ and $\delta\phi_2$ that are not compensated anymore. The shift of the maximum is established from the analytical expressions for the space and phase shifts acquired by the NLS solitons of unequal amplitudes (Novikov et al. (1984)).

The numerical simulations of the soliton interactions in the DZe equation were carried out in a periodic domain $x \in [0, 100\,\lambda_0]$. In order to study the influence of the relative phase on the value of the maximum amplification $A(\Delta\phi)$, a sequence of simulations was performed with various values of the initial phase $\phi_{01}$. By using the formulas (14) and (6), we find the dependence of the DZe soliton phase at its center on time:

$$\phi(t) = \phi_0 - (\Omega + k_0 V - \omega_{k_0})t. \tag{21}$$

The relative phase of the solitons having different parameters $\Omega_1$, $\Omega_2$ and velocities $V = V_0 \pm U_0$ is given by the following expression

$$\Delta\phi(t) = \phi_{02} - \phi_{01} - (\Omega_2 - \Omega_1)t + 2k_0 U_0 t, \tag{22}$$

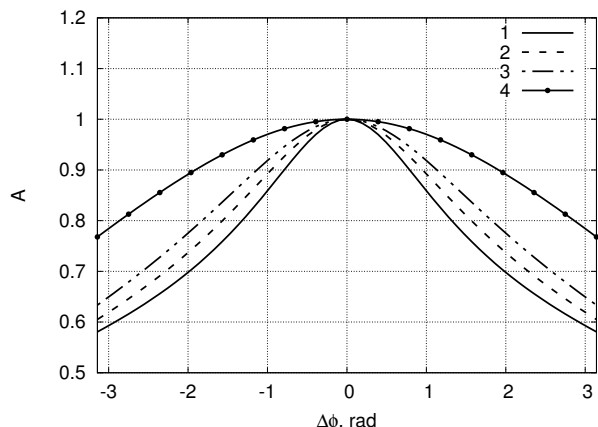

**Figure 3.** The maximum amplification $A$ of the wave field amplitude of colliding NLS solitons depending on the relative phase $\Delta\phi$. The values of the wave steepnesses (and amplitudes) of the solitons are the following: $\mu \approx 0.2$, $C_0 = 1.11 \cdot 10^{-1} \sqrt{\omega_{k_0}}/k_0$ (solid curve 1); $\mu \approx 0.15$, $C_0 = 8.85 \cdot 10^{-2} \sqrt{\omega_{k_0}}/k_0$ (dashed curve 2); $\mu \approx 0.1$, $C_0 = 7.12 \cdot 10^{-2} \sqrt{\omega_{k_0}}/k_0$ (dash-dotted curve 3); $\mu \approx 0.05$ $C_0 = 3.16 \cdot 10^{-2} \sqrt{\omega_{k_0}}/k_0$ (solid curve 4 with dots).

and is not time invariant. Thus, we define the phase difference of the solitons at the moment of time $t_c = 25\lambda_0/U_0 = 1250\,T_0$ as:

$$\Delta\phi = (\phi_{02} - \phi_{01}) - \frac{25\lambda_0(\Omega_2 - \Omega_1)}{U_0} + 50k_0\lambda_0, \tag{23}$$

In addition the solitons acquire space and phase shifts which cannot be simply accounted in $\Delta\phi$.

### 4.1 Soliton collisions: amplitude amplification and energy loss

Our numerical simulations show that the dependences $A(\Delta\phi)$ in the DZe equation and in the NLS equation are similar when soliton amplitudes are small ($\mu \approx 0.05$). In this case the maximum and the minimum of the amplitude amplification function for the DZe model are observed at $\Delta\phi \approx 0$ and at $\Delta\phi \approx \pm\pi$ respectively, i.e. similar to the NLS case – compare the solid curves with dots (curves 4) in figures 3 and 4. At larger values of the wave steepness the position of maximum of $A(\Delta\phi)$ is shifted from $\Delta\phi = 0$ more significantly – see the figure 4. We believe that the shift of the maximum of $A(\Delta\phi)$ can be compensated by choosing more precise definition of the soliton phase difference (23), that is a subject for further studies.

We found that the maximum value of amplitude amplification $A(\Delta\phi)$ increases with the initial amplitude (and steepness) of the DZe solitons and exceeds $A = 1$ by almost 20% for maximum value of the wave steepness studied in this work ($\mu \approx 0.2$) – see again the figure 4. As shown in the figures 3 and 4, the minimum value of the amplitude amplification for steep solitons $A \approx 0.6$, thus in this case the wave field slightly increases the soliton amplitudes. The envelope profiles of colliding DZe solitons having the wave steepness $\mu \approx 0.2$ are shown in the figures 5 and 6 for the values of the relative phase $\Delta\phi$ corresponding to the minimum and maximum amplification $A(\Delta\phi)$.

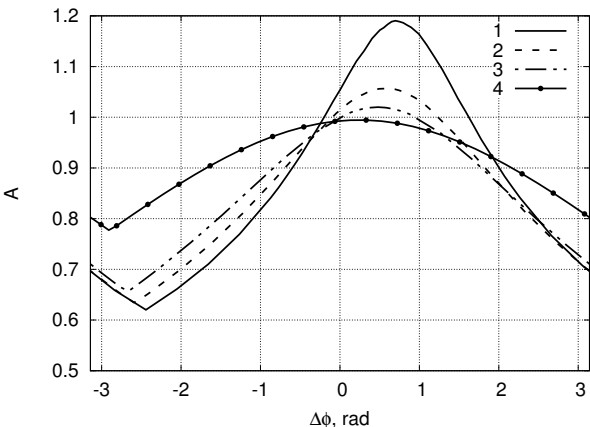

**Figure 4.** The maximum amplification $A$ of the wave field amplitude of colliding DZe solitons depending on the relative phase $\Delta\phi$. The values of the wave steepnesses (and amplitudes) of the solitons are the following: $\mu \approx 0.2$, $C_0 = 1.11 \cdot 10^{-1} \sqrt{\omega_{k_0}}/k_0$ (solid curve 1); $\mu \approx 0.15$, $C_0 = 8.85 \cdot 10^{-2} \sqrt{\omega_{k_0}}/k_0$ (dashed curve 2); $\mu \approx 0.1$, $C_0 = 7.12 \cdot 10^{-2} \sqrt{\omega_{k_0}}/k_0$ (dash-dotted curve 3); $\mu \approx 0.05$ $C_0 = 3.16 \cdot 10^{-2} \sqrt{\omega_{k_0}}/k_0$ (solid curve 4 with dots).

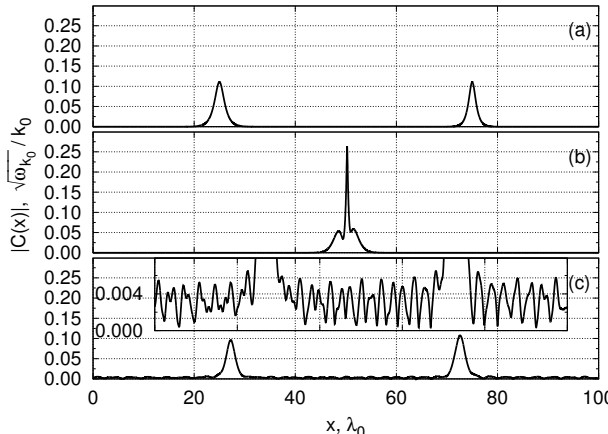

**Figure 5.** Collision of DZe solitons having the wave steepness $\mu \approx 0.2$ and the phase difference $\Delta\phi \approx 0.7$. Snapshots show the absolute value of the envelope function $|C(x)|$ at the initial moment of simulation $t = 0$ (snapshot (a)); at the moment of maximum amplitude amplification $t = 1214.35\,T_0$ (snapshot (b)) and at the final moment of simulation $t = 2500\,T_0$ (snapshot (c)). Zoom of the final amplitude profile is shown in the inset of the snapshot (c).

The interactions of solitons (or breathers) in the DZ model are inelastic (Dyachenko et al. (2013)), which is manifested by radiation of incoherent waves as can be seen from the figures 5 and 6. We have observed that level of the radiation is strongly dependent on the relative phase – compare the lower pictures in the figures 5 and 6.

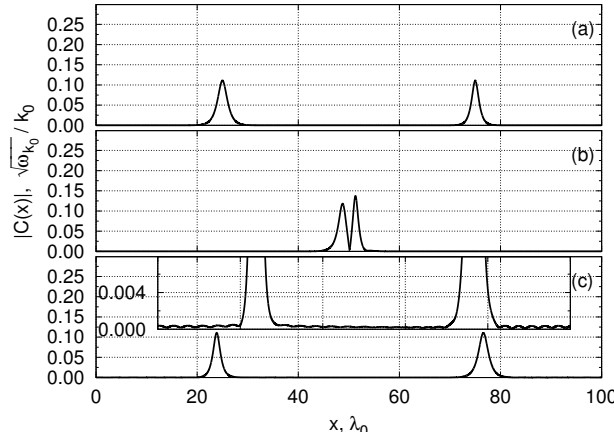

**Figure 6.** Collision of DZe solitons having the wave steepness $\mu \approx 0.2$ and the phase difference $\Delta\phi \approx -2.5$. Snapshots show the absolute value of the envelope function $|C(x)|$ at the initial moment of simulation $t = 0$ (snapshot (a)); at the moment of maximum amplitude amplification $t = 1193.66\,T_0$ (snapshot (b)) and at the final moment of simulation $t = 2500\,T_0$ (snapshot (c)). Zoom of the final amplitude profile is shown in the inset of the snapshot (c).

We quantitatively study the dependence of soliton energy losses $\Delta E_{loss}$ on the relative phase $\Delta\phi$. The total Hamiltonian of the wave field in the laboratory frame of reference is defined by the following expression:

$$H = \mathcal{H} + \frac{\omega_{k_0}}{2}N + \frac{\omega_{k_0}}{2k_0}P. \tag{24}$$

Here, the Hamiltonian $\mathcal{H}$ in the framework moving with the group velocity $V_0 = \frac{\omega_{k_0}}{2k_0}$ is defined by formula (12). $N$ and $P$ are
5   the number of waves and the horizontal momentum in the laboratory frame of reference:

$$\begin{aligned}
N &= \int\limits_{-k_0}^{\infty} \frac{|C_k|^2}{k_0 + k}\,dk, \\
P &= \int\limits_{-k_0}^{\infty} |C_k|^2\,dk. \tag{25}
\end{aligned}$$

Note, that the number of waves and the horizontal momentum are additional integrals of motion of the DZe equation (11). We denote the total energy of our system (i.e. the value of the Hamiltonian (24) at the whole spatial interval $[0, 100\lambda_0]$) as $E$, while the initial energies of the first and second soliton as $E_1$ and $E_2$. The values of energy change of each of the solitons
10   after collision we denote as $\delta E_1$ and $\delta E_2$. As mentioned above we mark the parameters of soliton initially located at $25\lambda_0$ by the index 1 and the parameters of soliton initially located at $75\lambda_0$ by the index 2. To estimate $\delta E_1$ and $\delta E_2$ we cut out each soliton after collision by a window function and then calculate the value of the Hamiltonian (24) for each part of the wave field. The window function was chosen so that being applied to a soliton propagating in the absence of another soliton allows us to estimate the value of the soliton energy with accuracy $0.01\%$. We define the total energy losses caused by the radiation

of incoherent waves relatively to the total energy of the system:

$$\Delta E_{loss} = -\frac{\delta E_1 + \delta E_2}{E}. \tag{26}$$

The figure 7 shows the energy losses as a function of the relative phase for the steepness of the colliding solitons $\mu \approx 0.2$. We have found that the value of the energy losses can reach $\approx 3\%$ at certain value of $\Delta\phi$. As one can see from the figures 4 and 7, the positions of maximum amplitude amplification and maximum of energy losses are strongly correlated.

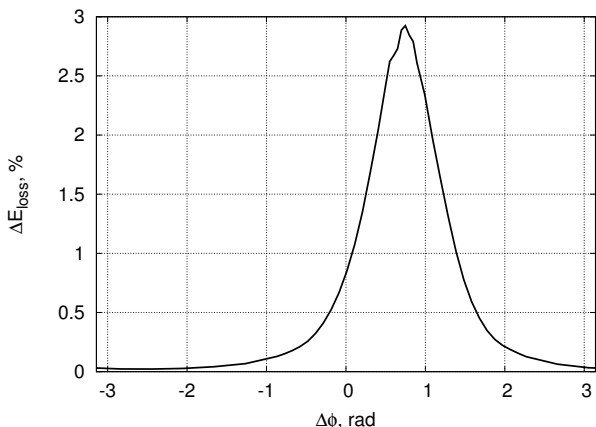

**Figure 7.** The total energy losses $\Delta E_{loss}$ (in percent – see formula (26)) of DZe solitons after their collision depending on the relative phase $\Delta\phi$. The wave steepness of the solitons $\mu \approx 0.2$.

### 4.2 Soliton collisions: space shifts and energy interchange

In this paragraph we describe the individual changes of DZe solitons after collision. We measure the energy changes of the soliton 1 and soliton 2 relative to their individual energies:

$$\Delta E_1 = \frac{\delta E_1}{E_1}, \quad \Delta E_2 = \frac{\delta E_2}{E_2}. \tag{27}$$

We have found that solitons of the DZe equation exchange energy with each other. Each of the solitons can gain or lose the energy after collision in dependence on the relative phase $\Delta\phi$ – see the figure 8. As one can see the maximum energy gain of the first soliton is achieved at $\Delta\phi \approx 1.5$, while the maximum energy gain of the second soliton is achieved at $\Delta\phi \approx 0$. It is interesting to note that the energy exchange between the solitons is absent at the values of the relative phase close to the maximum (i.e. at $\Delta\phi \approx 0.7$, see the figure 4) and to the minimum (i.e. at $\Delta\phi \approx 0.7 - \pi$, see the figure 4) of the wave field amplification. More precisely, at the point $\Delta\phi \approx 0.7$ we observe the intersection of the curves $\Delta E_1(\Delta\phi)$ and $\Delta E_2(\Delta\phi)$ – see the figure 8. In the intersection point $\Delta E_1 = \Delta E_2 = \Delta E'$ and thus $\Delta E_{loss} = -\Delta E'$ (see formulas (26) and (27)), i.e. the energy changes of each soliton are caused only by the radiation. The same situation is observed at the point $\Delta\phi \approx 0.7 - \pi$, but now the soliton energy loses are almost absent at all.

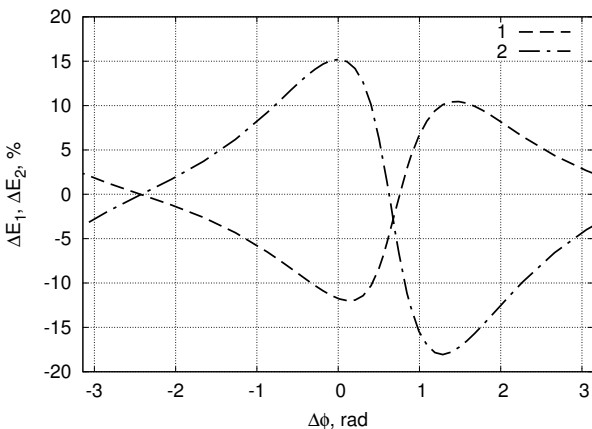

**Figure 8.** The individual energy change (in percent – see formula (27)) of DZe solitons after their collision depending on the relative phase $\Delta\phi$. The dashed curve 1 shows dependence of the energy change for the first soliton $\Delta E_1(\Delta\phi)$ while the dash-dotted curve 2 corresponds to dependence of the energy change for the second soliton $\Delta E_1(\Delta\phi)$. The wave steepness of the solitons $\mu \approx 0.2$.

The energy exchange and energy losses result in the increase or decrease of the soliton amplitudes, that is demonstrated by the figures 9(a) and 9(b). For each soliton we additionally simulate its propagation in the absence of the another soliton (i.e. in the absence of the interaction). In the figures 9(a) and 9(b) we show the envelope profiles of the solitons after collision in comparison with non-interacting solitons at the same moment of time ($t = 2500\,T_0$). The figure 9(a) corresponds to the relative

phase $\Delta\phi \approx 1.5$ and the figure 9(b) – to the relative phase $\Delta\phi \approx 0$.

In addition, the figures 9(a) and 9(b) demonstrate that the space positions of solitons after the interaction also depend on the relative phase $\Delta\phi$. We calculate the space shifts of the solitons $\delta x_1$ and $\delta x_2$ in the direction of soliton propagation (as we mentioned above we consider the system of reference moving with velocity $V_0$ where the solitons propagate in different directions) as difference in space positions between interacting and free propagating soliton at the same time $t = 2500\,T_0$.

We demonstrate the dependence of $\delta x_1$ and $\delta x_2$ on the relative phase $\Delta\phi$ in figure 10 for the values of the wave steepness $\mu \approx 0.05, 0.1, 0.15, 0.2$. In contrast to the NLS model the space shifts of solitons of the DZe equation can be either positive or negative at high values of $\mu$ – see the figure 10 (c,d). The curves $\delta x_{1,2}(\Delta\phi)$ become almost straight in the limit of small steepness (at $\mu \approx 0.05$ ) – see the figure 10 (a). We also show in the figure 10 the corresponding values of space shifts calculated using the NLS formula (17) for each value of $\mu$. Even at small steepness we observe a difference between soliton space shifts

in the DZe and NLS equations that we explain by the mentioned above difference between two-soliton wave groups in these two models.

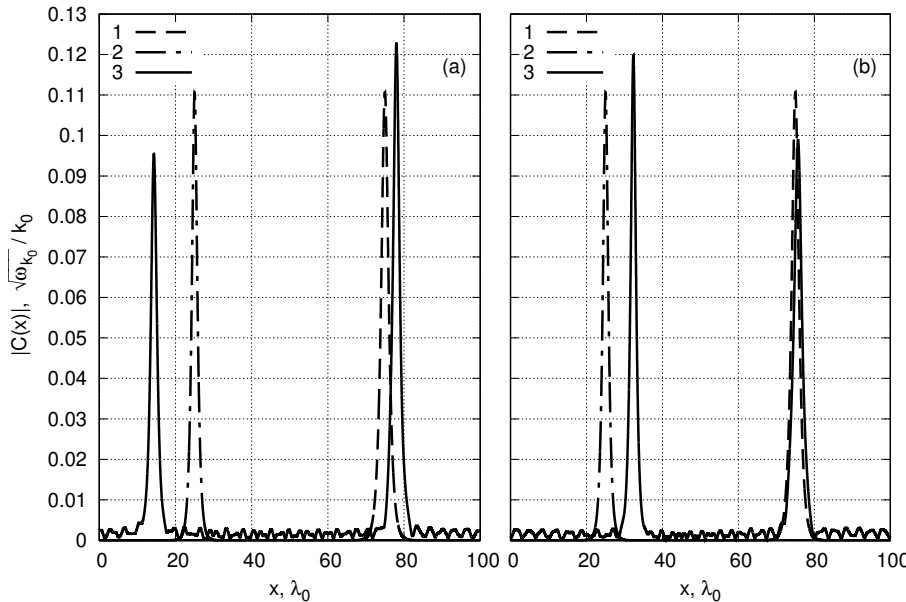

**Figure 9.** Comparison of the DZe solitons after mutual collision and the same solitons propagated without interaction. The wave steepness of the solitons $\mu \approx 0.2$ and the considered moment of time $t = 2500\,T_0$. The pictures (a) and (b) correspond to the relative phase of the colliding solitons $\Delta\phi \approx 1.5$ and $\Delta\phi \approx 0$ respectively. The solid curves 3 show modulus of the wave field amplitude $|C(x)|$ after collision of the first soliton having $U = 0.04\,V_0$ with the second soliton having $U = -0.04\,V_0$. The dashed curves 1 show the first soliton propagated in the absence of second soliton. The dash-dotted curves 2 show the second soliton propagated in the absence of the first soliton.

## 5  Conclusions

In this work we have studied how the relative phase of solitons in the DZe model affects the key properties of their interaction. All results presented here for solitons of the DZe equation are valid also for breathers of the DZ equation since solutions of these two models are linked by the transformation (10). In the first works devoting to numerical simulations of breather
5  interactions in the DZ equation (Fedele and Dutykh (2012a, b); Dyachenko et al. (2013)) the phase dependent effects were not studied and the wave steepness was taken to be small. For the chosen in the mentioned works breather phases and steepnesses a single collision of breathers does not lead to visible radiation of incoherent waves. However the minor energy radiation was registered after multiple breather collisions (Dyachenko et al. (2013)). Thus the breather interactions are not pure elastic that demonstrates nonintegrability of this model. The analytical proof of the nonintegrability of the DZ equation was also given in
10  the work of Dyachenko et al. (2013). Here we have studied the influence of the relative phase of the colliding solitons on the level of the radiation. We have found that the total energy loss due to the radiation is enhanced at a certain synchronisation of the relative phase between solitons. In this case the incoherent radiation becomes clearly visible even after a single collision – see the figure 5. We explain the latter in the following way. The maximum amplitude amplification is accompanied also by the

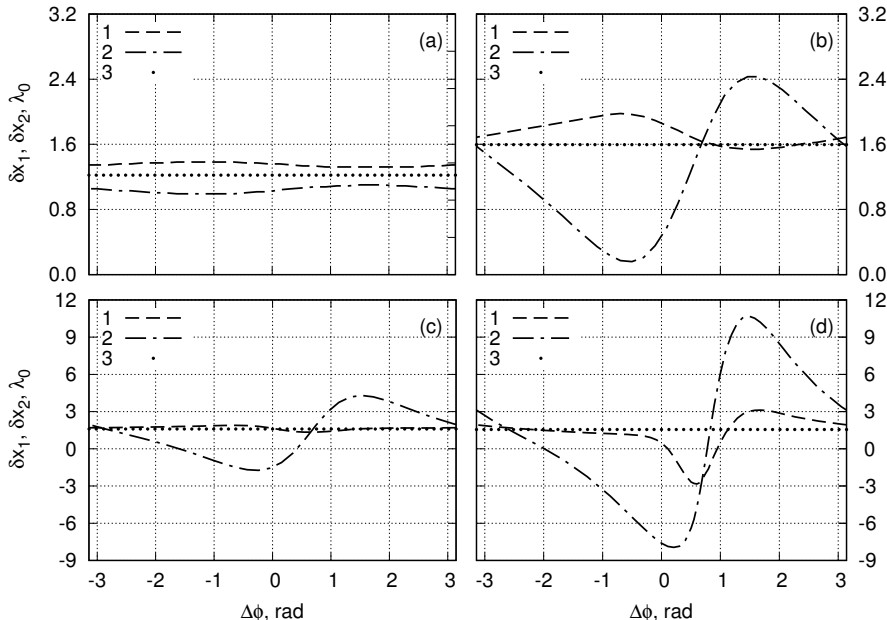

**Figure 10.** Space shifts of the DZe solitons depending on the relative phase $\Delta\phi$ for different wave steepnesses (and amplitudes): $\mu \sim 0.05$, $C_0 = 3.16 \cdot 10^{-2} \sqrt{\omega_{k_0}}/k_0$ (picture (a)); $\mu \sim 0.1$, $C_0 = 7.12 \cdot 10^{-2} \sqrt{\omega_{k_0}}/k_0$ (picture (b)); $\mu \sim 0.15$, $C_0 = 8.85 \cdot 10^{-2} \sqrt{\omega_{k_0}}/k_0$ (picture (c)); $\mu \sim 0.2$, $C_0 = 1.11 \cdot 10^{-1} \sqrt{\omega_{k_0}}/k_0$ (picture (d)). The dotted lines show the space shifts calculated in the frame of the NLS model via the analytical formula (17).

formation of the wave profile of high steepness. We have found that the maximum steepness reaches the value $\mu \approx 0.7$ during the collision process and thus the deviation of wave dynamics from the integrable model becomes to be significant.

Interactions of the breathers in the DZ equation at a certain phase synchronisation can lead to the formation of extreme amplitude waves. It is well known, that the maximum value of wave field as a result of soliton interactions in the NLS model
is equal to the sum of the soliton amplitudes. In this work we have found, that in the DZe equation the maximum amplification can be higher than the sum of amplitudes of the solitons. Interestingly, at large values of the wave field steepness this effect is enhanced, that can be a valuable complement in extreme amplitude waves studies.

We also have studied the phenomena of the energy exchange between the colliding solitons. This energy exchange is caused by inelasticity of the soliton interactions. The universal long term consequences of this process was studied in different noninte-
grable models (Krylov and Iankov (1980); Dyachenko et al. (1989)). It was shown that the numerous collisions and interactions with waves of radiation leads to formation of the powerful single solitary type wave (see the review by Zakharov and Kuznetsov (2012)). Here we have found that dynamics of a single collision is not universal: the direction of energy swap is determined by the soliton phases.

Furthermore, we have studied space shifts that solitons acquire after the collision. Soliton of the NLS equation always
acquire a positive constant shift $\delta x$ to its space position after interaction with other soliton moving with different velocity. The

value of $\delta x$ is defined only by the amplitudes and velocities of the colliding solitons. The interaction of solitons in the DZe equation also leads to the appearance of the space shifts. We show that the character of this effect is not universal ($\delta x$ can be positive or negative) and is determined in addition by the soliton phases.

The inelasticity of soliton collisions in nonintegrable models may destroy the initially coherent wave groups. However, as we have demonstrated here the total energy loss for interactions describing by the equation (1) does not exceed a few percent of energy of the solitons and we expect that observation of several subsequent soliton collisions is possible. The study of the influence of the relative phase of the colliding solitons in the fully nonlinear model is of fundamental interest. As was shown by Dyachenko et al. (2016b), the DZ equation quantitatively describes strongly nonlinear phenomena at the surface of deep fluid. Thus we believe that the effects reported here for the solitons of the DZe equation can be also observed for the fully nonlinear Euler equations.

Pairwise collisions of solitons (or breathers) is an important elementary process that can be observed in the wave dynamics of arbitrary disturbed fluid surface. For example, the recent numerical simulations of the DZe equation demonstrate that an ensemble of interacting solitons can appear as a result of modulation instability driven by random perturbations of an unstable plane wave (Dyachenko et al. (2017a)). Another important field of studies is the turbulence of rarified soliton gas where pairwise collision processes play the key role in the formation of wave field statistics (see the recent works of Pelinovsky et al. (2013); Shurgalina and Pelinovsky (2016)). We believe that results presented here can serve as a starting point in analytical description of such processes. Moreover, the reported dependence of soliton interaction dynamics on the relative phase is to be verified in laboratory experiments.

*Competing interests.* We declare that no competing interests are present.

*Acknowledgements.* The authors thank Dr. A.I. Dyachenko and Dr. S.A. Dyachenko for the helpful discussions. Theoretical part of the work was performed with support from the Russian Science Foundation (Grant No. 14-22-00174). The study reported in sections 4.1 and 4.2 (results of numerical simulations) was funded by RFBR and Government of the Novosibirsk region according to the research project No. 17-41-543347. Simulations were performed at the Novosibirsk Supercomputer Center of the Novosibirsk State University.

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
