# Peer review of "On the phase dependence of the soliton collisions in the Dyachenko-Zakharov envelope equation."

_Nonlinear Processes in Geophysics, 2018_

## Referee Comment (RC1) · Anonymous Referee #1 · 5 Mar 2018

The authors continue the work on soliton collisions of the compact Zakharov equation along the research line of Dyachenko et al.

They presented an analysis of soliton interactions studying phase shifts, energy losses/gains. The authors are not aware of past work on the compact Zakharov equation that provided new insights and highlights of this special equation.

The dynamics of the compact Zakharov equation is far richer than the NLS in terms of structures and dynamics in state space. In this regard, authors are unaware of previous work by Fedele & Dutykh (JFM, 2012), who revealed that existence of peakons of the compact Zakharov equation (cDZ) which bifurcate from smooth solitons. Moreover, the cDZ solitons loose energy after several collisions indicating some robustness properties of the cDZ.

Moreover, Fedele (JFM, 2014) was the first to derive a modified NLS equation from the compact ZakharoV equation, showing that the dynamics is that of an NLS for small wave steepness and it becomes of KdV type at higher steepness, suggesting a trend to wave breaking.

The authors have the great opportunity to study the richer structure of the compact Zakharov equation, beyond typical studies on NLS-type soliton collisions.

The dynamics of the compact Zakharov equation is far richer than the NLS and a much better approximation than the NLS to describe gravity water waves. It is about time to go beyond the NLS and explore the hidden richness of the cDZ dynamics

The authors have done a great job in studying the properties of cDZ soliton collisions as Dyachenko et al. have done previously.

I would like to see a major revised version of this work, where the authors take a steep further and go beyond typical NLS studies.

I suggest the authors to enrich their work by

1) analyzing the breaking of cdZ solitons and their bifurcation to peakons. 2) how peakons relate to the inelastic collisions of cDZ solitons ? Do solitons bifurcate to peakons before becoming inelastic ? 3) Why does it take several collisions before solitons radiate energy ?

REFERENCES

Fedele F. 2014 On certain properties of the compact Zakharov equation. J. Fluid Mech. 748, 692–711.

Fedele F. & Dutykh D. 2012 Special solutions to a compact equation for deep-water gravity waves. J. Fluid Mech. 712, 646–660.

---

## Referee Comment (RC2) · Anonymous Referee #1 · 30 Mar 2018

Dear Authors

Note that in Fedele (JFM, 2014) the projection operator P+ is discussed after Eq. 2.5 of that paper. The assumption of narrow-band envelope and/or large peak wavenumber k0 allows neglecting P+. Thus, the peakons found by Fedele and Dutykh (JFM, 2012) are asymptotically correct for asymptotically large peak wavenumber $k0 \gg 1$, suggesting a local structure (at small scales) of a breaking wave of the compact Zakharov equation.

Furthermore, irrespective of the P+ operator, the compact Zakharov equation manifests superharmonic instability above the critical steepness mu_c=0.577 (also proven for the

full Zakharov equation with no restrictions on wavenumbers, see paper by CRAWFORD et al. 1981). Thus, such instability is unaffected by P+ and it indicates a trend to breaking at high wavenumbers.

Note that both NLS and Dysthe do not manifest super harmonic instability. Thus, the compact Zakharov equation, with our without the P+ operator (Fedele & Dutykh 2012), goes beyond the NLS and Dysthe models as it has a built-in breaking-type mechanism at high wavenumbers. It may well be that the presence of the P+ operator may delay breaking to larger steepnesses (?!??).

Clearly, the P+ operator acts as a dissipation mechanism at low wavenumbers, and that's why peakons are not observed or one has to go to larger steep-nesses/wavenumbers to see them.

The two versions of the compact Zakharov equation, with or without the P+ operator, are both important for understanding wave breaking and the local structure of a breaker at high wavenumbers.

Said that, the revised manuscript is acceptable for publication.

References

CRAWFORD, D. R., LAKE, B. M., SAFFMAN, P. G. & YUEN, H. C. 1981 Stability of weakly nonlinear deep-water waves in two and three dimensions. J. Fluid Mech. 105, 177–191.

---

## Referee Comment (RC3) · Anonymous Referee #1 · 30 Mar 2018

Dear authors

I would like to see a discussion in your introduction about the compact Zakharov equation, with and without the P+ operator along the lines of my previous reply to you. I think it is important as it clarify the range of applicability of the two equations, with or without the P+ operator. Not only narrowband spectra, but also large peak wavenumber k0»1.

Note that in Fedele (JFM, 2014) the projection operator P+ is discussed after Eq. 2.5 of that paper. The assumption of narrow-band envelope and/or large peak wavenumber k0 allows neglecting P+. Thus, the peakons found by Fedele and Dutykh (JFM, 2012) are asymptotically correct for asymptotically large peak wavenumber k0Âz ÌǦ1,

suggest- ing a local structure (at small scales) of a breaking wave of the compact Za- kharov equation. Furthermore, irrespective of the P+ operator, the compact Zakharov equation manifests superharmonic instability above the critical steepness mu_c=0.577 (also proven for the full Zakharov equation with no restrictions on wavenumbers, see paper by CRAWFORD et al. 1981). Thus, such instability is unaffected by P+ and it indicates a trend to breaking at high wavenumbers. Note that both NLS and Dys- the do not manifest super harmonic instability. Thus, the compact Zakharov equa- tion, with our without the P+ operator (Fedele & Dutykh 2012), goes beyond the NLS and Dysthe models as it has a built-in breaking-type mechanism at high wavenum- bers. It may well be that the presence of the P+ operator may delay breaking to larger steepnesses (?!??). Clearly, the P+ operator acts as a dissipation mechanism at low wavenumbers, and that's why peakons are not observed or one has to go to larger steep- nesses/wavenumbers to see them. The two versions of the compact Zakharov equation, with or without the P+ operator, are both important for understanding wave breaking and the local structure of a breaker at high wavenumbers.

Said that, the revised manuscript is acceptable for publication.

References

CRAWFORD, D. R., LAKE, B. M., SAFFMAN, P. G. & YUEN, H. C. 1981 Stability of weakly nonlinear deep-water waves in two and three dimensions. J. Fluid Mech. 105, 177–191.

---

## Author Comment (AC1) · 30 Mar 2018

**Dear Editor and Reviewers,**

We agree, that the mentioned by the Referee1 works [1,2] are important in the studies of the properties of the deep water surface waves equations and we have added the corresponding short discussion to the section "1 Introduction" (see the corrected paragraph marked in red). However, we do not consider it appropriate to study peakon-type breathers in our work for the following reason.

First we note that in our work we study compact Zakharov equation for envelope written using "super compact" complex normal variable $c(x)$ (see the equation 1 in our manuscript). Meanwhile in the works [3,4,5] the compact Zakharov equation is written in different complex normal variable $b(x)$. In our response we use the notations from the papers [3,4,5], i.e. variable $b(x)$ and stress that the same arguments are also valid for the "super compact" variable $c(x)$.

The compact Zakharov equation is derived in the assumption that all waves are unidirectional [3,4,5]. This means that the Fourier spectrum of the normal variable $b(x)$ has only positive wavenumbers. Thus the dynamic equation includes projection operator $\hat{P}^+$ to the upper half plane. Eigenvalues of the operator $\hat{P}^+$ in the Fourier-space are presented by the step-function $\theta(k)$:

$$\begin{aligned} \theta(k) &= \quad 1, \quad if \quad k > 0 \\ \theta(k) &= \quad 0, \quad if \quad k \leq 0. \end{aligned}$$

The dynamic equation for the envelope $B(x,t)$ of the wave train with the carrier wave number $k_0$:

$$b(x,t) = B(x,t)e^{ik_0 x - i\omega t}, \tag{1}$$

must include the operators $\hat{\theta}(k_0+k)$ acting as $\theta(k_0+k)$ in nonlinear parts of the equation to cancel Fourier modes with the wavenumbers $k < -k_0$.

Unfortunately, the early papers of Dyachenko and Zakharov [3,4] have misprints. While the derivation of the compact Zakharov equation in the Fourier space is correct, in the final explicit form of the equation in x-space the projector $\hat{P}^+$ was missed (it was a typographical mistake). In the later papers (see, for instance [5]) this mistake was corrected.

In the paper mentioned by the Referee1 [1] the envelope equation has no operators of the form $\hat{\theta}(k_0 + k)$. Therefore, this equation is coincide with the compact Zakharov equation for envelope only in the case of the narrow spectral bandwidth $\frac{\Delta k}{k_0} < 1$. This is consistent with the fact that one can extract the NLS and the Dysthe equations both

from the Zakharov equation for envelope [6] and from the envelope equation written in the paper [1].

However without using of the projection operator the equation written in the work [1] and equation studied in our work differ significantly at large spectral widths of the solutions i.e. in the particular case of peakons. Indeed, as was found in the work [1], peakons have large spectral width – see the sentence at the page 652 "Figure 7(a) shows the numerically converged peakon obtained via the Petviashvili scheme using $N \sim 1.5 \times 10^6$ Fourier modes...". In our preliminary simulations that we have made with compact Zakharov equation (including the projection operator) using the Petviashvili method we do not observe the bifurcation of breathers to peakons even at high wave steepness. Our numerical Petviashvili scheme allows to find very narrow breathers having only three waves under the envelope. The wave steepness of such solutions reaches $\approx 0.6$. As we understand the results of the work [1] the bifurcation of breathers to peakons was observed already at significantly smaller values of wave steepness. Thus we believe that the question about peakon-type solutions in the compact Zakharov equation for envelope of unidirectional waves demands separate detailed study.

In our work we focus on the breather solutions that were recently successfully reproduced in the water wave tanks – see the works [7,8]. We stress, that we have found the fundamental differences of the breather interactions comparing to the simple NLSE soliton dynamics. We observe not only radiation and magnification of amplitude amplification, but also the energy exchange between breathers and nontrivial behaviour of the breather space shifts. All the mentioned effects turned out to be phase-dependent and is to be studied in laboratory experiments.

**Answer to the question "Why does it take several collisions before solitons radiate energy?"**

Indeed, previously Dyachenko, Kachulin and Zakharov shown in the work [3], that multiple breather collision in the compact Zakharov equation leads to the energy losses.

However in the work [3] only one particular case of breather phases was studied and the steepness of the breathers in numerical experiments was small, about 0.08. Thus, the minor energy radiation becomes clearly visible only after the 100 breather collisions. In our work we study the complete dependence of energy looses from the relative breathers phase in the case of lager breather steepness 0.15. We have found that the level of radiation reaches 3% of the total breather energy at certain phase synchronisation $\Delta\theta \approx 0$ (see the figure 7 in our manuscript), so that incoherent radiation is distinguishable even after a single interaction of breathters – see the last snapshot in figure 5.

We have added the corresponding short discussion to the section "5 Conclusions" (see the corrected paragraph marked in red).

**Yours sincerely,**

**Authors**

**References**

[1] Fedele F. and Dutykh D. (2012). Special solutions to a compact equation for deep-water gravity waves. J. Fluid Mech. 712, 646–660.

[2] Fedele, F. (2014). On certain properties of the compact Zakharov equation. Journal of Fluid Mechanics, 748, 692-711.

[3] A. I. Dyachenko and V. E. Zakharov, (2011) Compact Equation for Gravity Waves on Deep Water, JETP Letters, Vol. 93, No. 12, pp. 701–705.

[4] A. I. Dyachenko and V. E. Zakharov, (2012) A dynamic equation for water waves in one horizontal dimension, European Journal of Mechanics B/Fluids 32, 17–21.
[5] Dyachenko, A., Kachulin, D., and Zakharov V. E., (2013). On the nonintegrability of the free surface hydrodynamics, JETP letters, 98, 43–47.

[6] A.I. Dyachenko, D.I. Kachulin, V.E. Zakharov, Envelope equation for water waves, J. Ocean Engineering & Marine Energy, 3(4), 409-415 (2017)

[7] Slunyaev, A., Clauss, G. F., Klein, M., and Onorato, M. (2013). Simulations and experiments of short intense envelope solitons of surface water waves. Physics of Fluids, 25(6), 067105.

[8] Slunyaev, A., Klein, M., and Clauss, G. F. (2017). Laboratory and numerical study of intense envelope solitons of water waves: Generation, reflection from a wall, and collisions. Physics of Fluids, 29(4), 047103.

---

## Author Comment (AC2) · 30 Mar 2018

**Phase-dependent dynamics of breather collisions in the compact Zakharov equation for envelope**

Dmitry Kachulin[1] and Andrey Gelash[1,2]

[1]Novosibirsk State University, Novosibirsk, 630090, Russia
[2]Institute of Thermophysics, SB RAS, Novosibirsk, 630090, Russia

**Correspondence:** Dmitry Kachulin, d.kachulin@gmail.com

**Abstract.** We study collisions of the solitary type coherent wave structures – breathers in the nonintegrable Zakharov equation for envelope describing gravity waves on the surface of deep water. The numerical simulations of breather interactions revealed several fundamentally different effects when compared to analytical two-soliton solutions of the nonlinear Schrodinger equation. The relative phase of the breathers is shown to be the key parameter determining the dynamics of the interaction. We show that the maximum of the amplitude amplification can significantly exceed the sum of the breather amplitudes. The breathers lose up to a few percent of their energy during the collisions due to radiation of incoherent waves and in addition exchange energy with each other. The level of the energy loss increases with large values of the wave steepness, and also with certain synchronisation of breather phases. Each of the breathers can gain or lose the energy after collision resulting in the increase or decrease of the amplitude. The magnitude of the space shifts that breathers acquire after collisions depends on the relative phase and can be either positive or negative.

**1 Introduction**

In the complex dynamics of nonlinear waves on the surface of deep water, one type of processes is of special attention in the experimental and theoretical studies: the propagation of the coherent wave structures – solitons and breathers, and their mutual interactions. The exact model for the surface waves are the Euler equations which are rather complicated for analytical analysis and numerical simulations. At the same time approximate models for deep water surface waves demonstrate good agreement with experimental data and are widely used in fluid dynamics and geophysics.

The first-order weakly nonlinear model for the surface waves is the nonlinear Schrodinger (NLS) equation. The NLS model describes propagation of envelope of the quasi-monochromatic wave packets (Zakharov (1968)) and in the one-dimensional case is completely integrable via the inverse scattering transform (IST) method (Zakharov and Shabat (1972)).

As well known, solitons and breathers appear as a result of a balance between nonlinearity and dispersion. In the case of the NLS equation, the IST allows to find soliton solutions and describe their interactions analytically. The complete integrability leads to the remarkable properties of the interaction process: solitons of the NLS equation collide absolutely elastically. The next order nonlinear model for description of the wave envelope propagation, the so called Dysthe equation (Dysthe (1979)) – is not integrable. However, the existence of solitary type solutions in this model was also demonstrated by different analytical

and numerical approaches – see Akylas (1989); Zakharov and Dyachenko (2010). In this work we will use the term "breathers" to describe solitary type solutions of the Dysthe equations and the next order equations.

The NLS and Dysthe envelope models assume that the wave field steepness $\mu$ is small and the wave train is weakly modulated, i.e. its spectrum belongs to a narrow vicinity of some carrier wave number $k_0$. The compact Zakharov equation is written for the wave train itself and free from the latter restriction (Dyachenko and Zakharov (2011, 2012)). Thus the compact Zakharov equation is able to describe the dynamics of nonlinear wave fields having broad spectrum and high steepness. Breather solutions of the compact Zakharov equation can be found numerically (Fedele and Dutykh (2012a, b); Dyachenko et al. (2013)). The first numerical simulations of breather interactions in the compact Zakharov equation were performed by Fedele and Dutykh (2012a, b) and by Dyachenko et al. (2013). The detailed analysis of the properties of the compact Zakharov equation at different wave steepness $\mu$ was performed in the work of Fedele (2014). In particular it was shown that the dynamics of the compact Zakharov equation becomes of a modified Korteweg–de Vries equation type at higher steepness providing a possible mechanism of wave breaking. Breather solutions can be also found numerically in the fully nonlinear Euler equations (Dyachenko and Zakharov (2008)). Recently, the propagation and interactions of the 
[revised manuscript text omitted]

---

## Referee Comment (RC4) · Anonymous Referee #2 · 1 Apr 2018

**Review**
on the manuscript by D. Kachulin and A. Gelash
"Phase-dependent dynamics of breather collisions in the compact Zakharov equation for envelope" submitted for publication in journal "Nonlinear Processes in Geophysics".

The manuscript represents the results of numerical study of collisions between soliton-like wave groups, and how the collision consequences depend on the phase difference between the colliding groups. The simulated equations are beyond the leading-order approximation provided by the nonlinear Schrodinger equation. The problem statement is generally understood; the results have rather fundamental significance and may be interesting for many readers bearing in mind that the soliton-like groups of steep waves have been recently measured in laboratory facility. At the same time I assume that in the present form the paper is not 'reader-friendly' and should be improved. Supposing that many of the readers of NPG deal with geophysical problems, the style of the paper is too much remote from practice. The authors forget to reformulate the obtained results in plane words to be perceived by a broad audience. This is the main my concern; I suggest to rewrite the text appreciably according to the particular suggestions below. I assume that after the revision the text may be published in NPG.

1. I would like to argue with the terminology used in the paper. It is difficult to understand why the solitary groups of waves within the NLS equations are *envelope solitons*, but in a generalized framework they are *breathers*. This difference is especially difficult to catch looking at the ansatz for *breathers* for the solution Eq. (3), which coincides with the one for *envelope solitons* of the NLS equations. Furthermore, it becomes puzzling if one recalls the *Peregrine breather*, *Akhmediev breather* etc, which are wave patterns of absolutely different style than the ones discussed in this work. The latter nomenclature has been already established and seems to be conventional (see e.g. the series of laboratory simulations by A. Chabchoub at al., well represented in the recent literature).

2. In the text there is a mixture of dimensional and non-dimensional quantities. Eq. (14) is dimensional, but $k_0 = 100$ and $g = 1$ – are not (just two lines above). All the results are expressed through unnaturally scaled variables ($C_0$, $U$, $\Omega$, $x$). They should be replaced by the properly scaled, i.e. $k_0C_0$, $k_0U/\omega_0$, $\Omega/\omega_0$, $k_0x$ or similar.

3. Interactions of soliton-like groups were studied numerically in the frameworks of the full Euler equations in the papers V.E. Zakharov, A.I. Dyachenko, A.O. Prokofiev, Eur. J. Mech. B/Fluids 25, 677 (2006) and A.V. Slunyaev, J. Exp. Theor. Phys. 109, 676 (2009). The approach to investigation of the soliton turbulence with the help of exact solutions (in integrable systems) was applied in the papers E.N. Pelinovsky, E.G. Shurgalina, A.V. Sergeeva, et al., Phys. Lett. A 377, 272 (2013) and E. Shurgalina, E. Pelinovsky, Phys. Lett. A 380, 2049 (2016). These references may be useful for the authors.

4. The figures are not suited for the black and white printing, though they may be easily improved if different line styles are used.

5. Page 2, lines 27-28: it should be clarified that the function of the velocity potential is defined on the water surface.

6. Page 3, line 6: should read "can be recovered".

7. Page 3, 4: The role of parameters $V$ and $\Omega$ should be better explained from the physical point of view. Namely, the natural physical parameters of the solitary group are its amplitude, carrier wave length, carrier frequency and the velocity. The carrier wavenumber, $\widetilde{k}$, is fixed equal to 100. The wave group velocity $V$ is a function of $\widetilde{k}$ and the group amplitude. Parameter $\Omega$ is the frequency in the reference moving with velocity $V$, which also depends on the wave amplitude.

8. The same comment to the beginning of Sec. 4: It is more natural to say that the wavenumber $k_0 = 100$ is fixed first, which results in the value $V_0 = 0.05$.

9. Page 4: When the equation on the envelope, Eq. (11), is introduced, the limits of its validity remain unclear. It is stated that the equations (1) and (11) equally accurate. The parameter $\widetilde{k}$ denotes the wavenumber of the soliton-like group, though $k_0$ in Eq. (11) is the parameter of this equation. If Eq. (11) is valid for any choice of $k_0$ (since no extra assumptions on the applicability of Eq. (11) implied), how the choice of $k_0$ results in the solution of the equation? Besides, please indicate what $\omega_{k0}$ exactly means.

10. The end of Sec. 2: it should be said clearly that the solutions of Eq. (11) and Eq. (13) will be compared in what follows.

11. Eq. (14): $x_0$ – is the solution location at $t = 0$. The notation for the wave amplitude with letter $C_0$ does not seem to be a good choice, having in mind that it is a typical notation for velocity.

12. Page 4, lines 25-26: The discussion of the Fourier modes for $C_s$ will be easier to understand if the formula for $C_{sk}$ is given. I suggest doing so. It is instructive to say in words that the shape and the width of the solitary group do not depend on $k_0$ and $V_0$, and are functions of amplitude only.

13. Page 5, line 1: It is unnatural to claim that the velocity $U$ leads to the wavenumber shift. Quite the opposite, the wavenumber offset $\Delta k$ leads to the velocity correction $U$ to the celerity $V_0$, so that the total group velocity is $V = V_0 + U$.

14. Page 5: "To obtain solitons of almost the same characteristic width as breathers we should vary the carrier wave number $k_0$ instead of the relative velocity $U$ in the solution (14). But this procedure is not appropriate for our work since in this case we are not able to use the NLS equation to study soliton interactions" – this discussion may be correct from the technical point of view, but it sounds senseless from the physical point of view. The point is that the group shapes for the given amplitude are different in the frameworks of the NLS equation and the compact Zakharov equation. The groups with same characteristic widths will have different amplitudes.

15. I have a simple reasoning which explains qualitatively the curves in Fig. 1. The actual wavenumber $k_s = k_0 - \Delta k$ is larger when $U < 0$, thus for the given amplitude $A_s$ the wave will be steeper (steepness $k_s A_s$ controls the nonlinearity). Accordingly, steeper waves need stronger dispersion to balance the soliton, and hence the group is shorter when $U < 0$.

16. Sec. 4: It will be useful to say in words that you simulate two soliton-like groups with the same amplitude which travel in the same direction with close velocities $V_0 + U_0$ and $V_0 – U_0$; the consideration will be given for the reference moving with velocity $V_0$.

17. Page 6, lines 10-11: It is important to clarify that the "each of solitons acquires a positive shift in the direction of soliton propagation" when considered in the system of reference moving with velocity $V_0$, thus the solitons propagate in different directions in this reference.

18. Eq. (16): the signs in the formula seem to be inconsistent with Eq. (14) (at the second term in (16)). It may be constructive to comment that the phase corrections in Eq. (16) are related to the nonlinear frequency shift (the latter term) and to the offset of the carrier wavenumber (it is probably better to rewrite the corresponding summand in terms of $\Delta k$).

19. Page 6, lines 18-20: The formula for the phase shift should be given. As far as I understood, the 'additional phase shifts, which the solitons acquire' are not *neglected*, but are mutually *compensated*. This issue will be clear if the mentioned formula is provided.

20. Page 6, lines 25-26: The phrase "in the general case of different solitons" is not accurate. It is better to say clearer that when the solitons are characterized by different amplitudes or/and different velocity shift parameters $|U|$.

21. Page 7, line 4: "The colliding breathers have slightly different widths, so their relative phase is not time invariant." It is not clear how different widths make the relative phase varying. Please clarify.

22. Page 8, line 7: The shift of the location of maximum of $A(\Delta\phi)$ may be due to the improper choice of the definition (19) of the relative phase. If the blue curve in Fig. 4 is centered, in the intervals of large $|\Delta\phi|$ figures 3 and 4 look qualitatively similar.

23. Page 8, line 13: "as can be seen from the figures 5 and 6." This phrase may be applied to Fig. 5 only.

24. Eq. (20): Please give the definition for the 'energy', and clarify what is the relative energy loss and exchange (relative to the total energy or to the individual soliton energy, etc).

25. Fig. 8: It seems that when $\Delta\phi = 0.58$ the total energy in the system increases after the collision. How could it happen?

26. Page 11, line 16: "We have found that the total energy loss due to the radiation is enhanced at large values of the wave steepness" – this statement was not discussed in the text, no proof of this fact was provided.

27. It seems that Fig. 9 and 10 are not consistent with Fig. 8. According to Fig. 8, the maximum amplification should be obtained by the second soliton when $\Delta\phi = 0.58$. However, the amplification of the second soliton seems to be larger in Fig. 9 compared to Fig. 10.

28. Fig. 11: the figure looks somewhat surprising. I cannot imagine how the curves tend to the limit of small wave amplitudes, when they should collapse to a single straight line. Could you please comment on this.

29. Page 12, lines 4-5: The phrase "the maximum value of the amplitude amplification … is equal to the sum of the soliton amplitudes" does not sound well. The maximum *value* of the field is equal to the sum of the soliton amplitudes.

30. Page 13, line 13: It is better to clarify the sentence as follows: "and is determined *in addition* by the breather phases."

31. I found some faults in English, they may be more. In some places they use 'to depend from' instead of 'to depend on' (page 6, lines 14, 22, caption of Fig. 4); 'can be power by' instead of 'can be powered by' (page 3, line 6); "shown" instead of "showed" (page 11, line 12). Please proofread carefully.

---

## Author Comment (AC3) · 30 Apr 2018

Dear Editor and Referees,

In general, we agree with the comments of the Referee2 and we believe that the suggested corrections will improve the presentation of the work. Now we are working on the manuscript. We plan to complete all the changes within three weeks.

Yours sincerely,

Dr. Dmitry Kachulin and Dr. Andrey Gelash

---

## Author Response (AR1)

Dear Editor and Reviewers,

The main changes in the revised version of our manuscript are the following. Firstly, we have taken into account the comment (1) of the Referee 2 about the terminology we use. Now we use the term soliton for solutions of the equation for envelope and the term breather for solutions of the compact equation written for the wave train itself (see also our response to the comment (1) below). In addition, we have revised the terminology for the water wave equations. The compact version of the Zakharov equation was derived by Dyachenko and Zakharov in 2011. The envelope version of Zakharov equation was presented by [Dyachenko et al. (2017a)] in the last year. So, the conventional terminology for these versions of Zakharov equation is not established yet. It was suggested in the work [Fedele, F., & Dutykh, D. (2012)] to denote the compact version of the Zakharov equation as compact Dyachenko-Zakharov (DZ) equation. We believe that this nomenclature is better that we used in the previous version of our manuscript. So, the envelope version of the Zakharov equation we now refer to as the Dyachenko--Zakharov envelope equation, or the DZe equation.

We also have extended our study of the soliton space shifts. In accordance with the question (28) of the Referee 2 we performed the additional numerical simulations for the wave steepness: 0.05, 0.1, and 0.15. We have demonstrated how the dependence of the space shifts on $\Delta\phi$ changes with the value of $\mu$ - see the figure 10 and also our comment to the question (28).

Then we have found an inaccuracy in the calculation (caused by a misprint in our numerical code) of the origin and sign of $\Delta\phi$. After the appropriate correction, all curves depended on $\Delta\phi$ have been mirrored and shifted a little to the right – see for example the figure 4. We apologise for this inaccuracy and confirm that it does not affect the results of the work. To demonstrate that in the limit of small steepness the dependence of the amplitude amplification becomes very similar to those in the NLS case, we have added simulations for the steepness $\mu = 0.05$ (compare the curves marked as 4 in the figures 3 and 4).

Finally, we have taken into account the comment (2) of the Referee 2 and now present all results of our numerical simulations in scaled variables – see also our response to the comment (2).

1. The term breathers now is conventional for solitary wave group solutions of the compact Dyachenko-Zakharov (DZ) equation (see the works of Dyachenko, Kachulin and Zakharov cited in the manuscript). We prefer to remain this nomenclature unchanged. However, we agree with the Referee 2, that in case of the envelope Dyachenko-Zakharov (DZe) equation the terms soliton and solitary wave group are more relevant. We have changed the terminology in the whole manuscript and added a short discussion about soliton and breather terminology to the "Introduction" section - see the text marked in red on the pages 1 and 2.

2. Thank you for this question. Now we use the following scaled variables when we present results of the simulations: $\lambda_0 = \frac{2\pi}{k_0}$ (unit of length), $T_0 = \frac{2\pi}{\omega_{k0}} = \frac{2\pi}{\sqrt{g\,k_0}}$ (unit of time). We measure the velocity of solitons in the unit of group velocity $V_0 = \frac{\omega_{k0}}{2k_0}$ corresponding to the characteristic wavelength. The dimensionless wave steepness $\mu \sim C_0 k_0 / \sqrt{\omega_{k0}}$, that is why we measure the wave field amplitude $C$ in the units $\sqrt{\omega_{k0}}/k_0$.

3. We agree that the suggested references are important and relevant. We have added them (with a sort discussion) to the "Introduction" section (page 2) and to the end of the "Conclusions" section (page 15) – see the text marked in red.

4. We have revised all our figures. Now we use different line stiles suitable for black and white printing. The style of the coordinate axes was also improved.

5. Done, marked in red on the page 3.

6. Fixed.

7. We have explained the meaning of the parameters $V$ and $\Omega$ in details on the page 4 – see the text marked in red. We stressed that $\Omega$ is nothing more than the new notation for the breather (as well as for the DZe soliton) frequency parameter – see the formula (6). The breather (as well as DZe soliton) solution is determined by two independent parameters: the group velocity $V$ in the laboratory frame of reference and the frequency $\Omega$. Note, that the carrier

wavenumber of the breathers (DZe solitons) $\tilde{k}$ is not fixed equal to 100. The value of the solitary group carrier wave number $\tilde{k}$ (and the carrier wave length $\lambda = 2\pi/\tilde{k}$ ) is defined by the parameter $V = \frac{1}{2}\sqrt{g/\tilde{k}}$. The second parameter $\Omega$ has the value close to $\frac{\sqrt{g\,\tilde{k}}}{2}$ (or $g/4V$ see formula (6)) and implicitly defines the shape and the amplitude of the breather (DZe solitons).

8. We agree with this comment. We have added one additional sentence to the beginning of the section 4 clarifying that we fix $k_0$ first. Note that now we use scaled variables according to the comment (2).

9. The equation (1) and the DZe equation (11) have the same accuracy. The DZe equation (11) was derived without any assumptions about spectral width of the wave field – see our comment marked in red after formula (13). The DZe equation is valid for any choice of $k_0$. The choice of $k_0$ do not affect to the breather solution of the equation (1). We added the definition of $\omega_{k0} = \sqrt{gk_0}$ to page 4 (red text after the formula (10)) and clarified that $k_0$ is an arbitrary wavenumber.

10. Done, marked in red on the page 5.

11. We have clarified the meaning of $x_0$ at the page 5, right after the formula (16). However, we prefer to remain our notation $C_0$ for the soliton amplitude, since in the case of DZe equation the letter $C$ is conventional notation for the wave field amplitude.

12, 13, 14. Thank you for these comments. We agree that the section 3 in its original version included many technical details which made the text not clear. We have excluded several sentences and now simply write that the group shapes for the given amplitude are different in the frameworks of the NLS equation and DZe equation (as was suggested by the Referee 2). We hope that the section 3 became self-consistent. We also believe that now the expression for $C_{sk}$ is not necessary since we do not discuss the shift of the soliton carrier wave number anymore.

15. Thank you for this useful comment. We have added this explanation to the end of the section 3 – see the text marked in red.

16. Done, marked in red at the beginning of the section 4.

17. Done, marked in red on the page – see the text marked in red before the formula (17).

18. No, everything is correct. The phase dependence $\phi(x,t)$ (see the formula (16)) is given by
$$\phi(x,t) = \phi_0 - \frac{4k_0^2}{\omega_{k_0}}U(x - x_0) + \frac{2k_0^2}{\omega_{k_0}}Ut - \frac{C_0^2 k_0^2}{2}t.$$
The time dependence $\phi(t)$ for the moving soliton ($x = x_0 + Ut$):
$$\phi(x,t) = \phi_0 - \frac{2k_0^2}{\omega_{k_0}}Ut - \frac{C_0^2 k_0^2}{2}t.$$

19. We have added the formula (18) which describes the phase shift in the case of NLS solitons. The sentence after formula 19 was also corrected – see the text marked in red.

20. Done, marked in red on the page 8.

21. We agree with this comment. We have added the formula (22) which clarify that the difference in parameters $\Omega_1$ and $\Omega_2$ makes the phase not time invariant.

22. This question is very important. Indeed, we use the simplest definition of the relative phase $\Delta\phi$ given by formula (23). The expression (23) allows us only to compensate the phase difference that solitons acquire during propagation to the collision area. We agree that the more appropriate choice of the definition of $\Delta\phi$ is needed – see our comment marked in red at the begging of the section 4.1. We have tried to account phase shifts, changes in the propagation time due to space shifts and some other effects to correct the definition of $\Delta\phi$. However, we have no better variant for the expression (23) so far.

23. The scale of the inset picture in the figure 6 was not appropriate to represent the low amplitude radiation. We have improved the inset pictures in the figures 5 and 6 to make the radiation visible.

24 and 25. We have added the expression for the Hamiltonian in $x$-space (formula (12)) and clarified at the page 11 that it defines the energy of the wave field (see the formula (24) and the red text around it). In addition, we have clarified that we calculate the total energy loss relative to the total energy of our system (see formula (26) and the red text above), while the individual changes of soliton energies are calculated relative to their individual energies (see the beginning of the section 4.2 and formula (27)). The latter means that in the figure 8 each curve is normalized on different value of soliton energy and thus the sum of the presented functions is not constant.

26. We removed the part of the statement "…the total energy loss due to the radiation is enhanced at large values of the wave steepness…" and hope that now the mentioned sentence is self-consistent – see the text marked in red in the "Conclusion" section.

27. We believe that the problem was caused by the non-clear explanation of which soliton we mark as first and which soliton we mark as second. We have stressed at the beginning of the section 4 and than recall one more time on the page 11 that the soliton 1 is initially located at the left and the soliton 2 is initially located at the right. After collision the solitons swapped their places, so the second soliton in the figure 9 is located at the left. We have checked that the figure 8 is consistent with the figure 9.

28. Thank you for this question. We have performed additional numerical simulations for the wave steepness: 0.05, 0.1, and 0.15. The results are presented in figure 10. We hope that now it is clear how the soliton space shifts curves tend to the small steepness limit. The curves become almost straight, however there are still two separate curves due to the discussed in the manuscript differences between solitons in the NLS and DZe models. We have added the additional discussion of the figure 10 at the end of the section 4.2. – see the text marked in red.

29. Done, marked in red in the "Conclusion" section.

30. Done, marked in red in the "Conclusion" section.

31. We have edited the whole text of the manuscript and improved it significantly.

---

## Referee Report (RR1)

**Review**
on the revised manuscript by D. Kachulin and A. Gelash
"On the phase dependence of the soliton collisions in the Dyachenko-Zakharov envelope equation" submitted for publication in journal "Nonlinear Processes in Geophysics".

The paper has been improved significantly; my comments have been taken into account. Unfortunately, I am not fully convinced by the authors' reply on the difference between solitons and breathers in their terminology, but leave this issue at the authors' discretion.

I still have a few more critical remarks which should be taken into account before the text may be published.

**1)** Page 1, lines 22 and below. The paragraph is not logical. In the discussion of the NLS and Dysthe equations it is crucial to emphasize that the integrable NLS equation possesses the mathematically strict soliton solution (i.e., with elastic collisions). While other nonintegrable equations may still have exact stationary solitary solutions (~"solitons"), which do not interact elastically.

The first sentence ("A term soliton was originally coined for a special solution of the NLS...") is not at the right place and should be shifted to the end of the paragraph, or may be to one of the subsequent paragraphs. The term 'soliton' was first attributed to the solutions of the KdV equation, hence the sentence sounds confusing.

**2)** Page 2, line 7: "The DZ equation is formulated for the wave train itself" – As I may understand, in this sentence and above the authors wish to oppose the NLS and Dysthe equations for the *modulation* or *envelope* against the DZ equation for the surface displacement and surface velocity potential. In the present form it is not clear. I suggest the following redaction:

*"Both the NLS and the Dysthe equations are formulated to describe the evolution of the envelope function. They require that the steepness of the wave train is small and it is modulated weakly, i.e., there are sufficiently many carrier wave lengths in the characteristic scale of the modulation. In terms of the Fourier transform of the surface elevation this is equivalent to having a sufficiently narrow band concentrated in the vicinity the carrier wave number. The DZ equation is written in terms of the surface displacement and the surface velocity potential, and is free from the assumptions of the weak nonlinearity and narrow bandness (Dyachenko and Zakharov (2011, 2012))."*

**3)** Page 9, line 14: You may note that when the solitons are steep, the maximum amplification increases by about 20%, and at the same time the minimum amplification decreases significantly. So that if the phases of two steep colliding solitons are chosen properly, the wave field may increase very little: $A \approx 0.6$ (see the solid curve in Fig. 4, and also Fig. 6; $A = 0.5$ for non-colliding solitons).

**4)** Page 14, lines 5-6: The sentence "All results presented here for solitons of the DZe equation are valid also for breathers of the DZ equation since these two models are physically identical" is very important as it describes the applicability limits of the study, but it is unclear. What does 'physically identical' mean? Are solutions of the DZ equation necessarily solutions of the DZe model? Is the opposite statement correct? The new title of the manuscript mentions the DZe model, not the DZ equation as before...

**5)** Page 15, line 22: "...by the absence of exact N-soliton solution formulas, and also the inelasticity of the interaction..." – inelasticity of the interaction guarantees the absence of the exact N-soliton solution. Thus, the two listed reasons are not of a similar weight.

There are also some drawbacks of the technical matter:

6. page 1, line 17: the name Schrödinger should be corrected.
7. page 2, line 5: "...on the characteristic **wave** length scale of the envelope modulation." should be replaced by "...on the characteristic length scale of the envelope modulation."
8. page 2, line 6: double use of "the" before "Fourier spectrum"
9. page 2, lines 31-32: "...including mKdV equation for **shallow water** waves..." should be replaced by "...including mKdV equation for **long** waves...", as the mKdV equation is a **long-wave** model, but generally speaking it does not describe **shallow water waves**.
10. page 5, line 16-17: It is better to make the sentence "The amplitude of the DZe soliton $C_0$ is not an independent parameter of the solution." more precise as following, "The amplitude of the DZe soliton $C_0$ is not a free parameter of the solution when $k_0$ and $V$ are fixed."
11. page 5, line 4: To change "$\delta x$ is $1.55\lambda_0$" to "$\delta x = 1.55\lambda_0$"
12. page 5, line 10: To change "For this case..." to "For the case of the NLS equation..."
13. page 14, line 1: To change "explain by the mention above" by "explain by the mention**ed** above"

---

## Author Response (AR2)

Dear Editor and Reviewers,

We have taken into account the additional recommendations of the Referee 2. The changes in the manuscript are marked in red and listed below (in accordance with the report of the Referee).

1)      We have added one sentence to the Introduction section page 1, that as we believe makes the paragraph more coherent. We also agree, that the sentence about the term "soliton" is confusing and we have decided to remove it from the manuscript.

2)      We agree, that our phrase "formulated for the wave train itself" is not clear. From the other side, the redaction suggested by the Referee is also not completely correct since the DZ equation is written in terms of special canonical variables. Thus we have used the text suggested by the Referee with some additional corrections – see the page 2.

3)       We have added a short discussion of the minimum value of the amplitude amplification to the section 4, page 8 and to the beginning of the section 4.1 - see the text marked in red.

4)      The DZe equation is the envelope form of the DZ equation. Thus the DZe equation has the same range of applicability and solutions of the DZ and DZe equations are in one-to-one correspondence. We have clarified it on the page 5, right after the formula (13) – see the text marked in red, see also the sort text in red at the beginning of the Conclusion section.

5)      We agree with this comment. It is non necessary to say about the absence of exact N-soliton solutions in this paragraph. We have remained only the statement about the inelasticity of the interactions – see the corrected text marked in red on the page 16.

The simple technical corrections 6-9 and 11-13 were made and marked by the red colour. In accordance with the comment 10 we have clarified that the problem with amplitude parameter C0 is the absence of analytical relation between C0 and the input parameters of the Petviashvili method - V and Omega. Thus to find DZe soliton having certain C0 and V, we fix V and vary Omega – see the text marked in red at the beginning of the section 3.

Sincerely yours,
Dr. Dmitry I. Kachulin and Dr. Andrey A. Gelash